# Molecular Mechanisms Associated with Antifungal Resistance in Pathogenic *Candida* Species

**DOI:** 10.3390/cells12222655

**Published:** 2023-11-19

**Authors:** Karolina M. Czajka, Krishnan Venkataraman, Danielle Brabant-Kirwan, Stacey A. Santi, Chris Verschoor, Vasu D. Appanna, Ravi Singh, Deborah P. Saunders, Sujeenthar Tharmalingam

**Affiliations:** 1Medical Sciences Division, NOSM University, 935 Ramsey Lake Rd., Sudbury, ON P3E 2C6, Canada; kczajka@nosm.ca (K.M.C.); kvenkataraman@laurentian.ca (K.V.); cverschoor@nosm.ca (C.V.); rsingh@hsnsudbury.ca (R.S.); dsaunders@hsnsudbury.ca (D.P.S.); 2School of Natural Sciences, Laurentian University, Sudbury, ON P3E 2C6, Canada; vappanna@laurentian.ca; 3Health Sciences North Research Institute, Sudbury, ON P3E 2H2, Canada; dbrabantkirwan@hsnsudbury.ca (D.B.-K.); ssanti@hsnri.ca (S.A.S.)

**Keywords:** antifungal, resistance, *Candida*, azoles, polyenes, echinocandins, fluorouracil, gene mutations, fungal infection, mutations, minimal inhibitory concentration, ergosterol, cell wall, efflux pumps, transporters

## Abstract

Candidiasis is a highly pervasive infection posing major health risks, especially for immunocompromised populations. Pathogenic *Candida* species have evolved intrinsic and acquired resistance to a variety of antifungal medications. The primary goal of this literature review is to summarize the molecular mechanisms associated with antifungal resistance in *Candida* species. Resistance can be conferred via gain-of-function mutations in target pathway genes or their transcriptional regulators. Therefore, an overview of the known gene mutations is presented for the following antifungals: azoles (fluconazole, voriconazole, posaconazole and itraconazole), echinocandins (caspofungin, anidulafungin and micafungin), polyenes (amphotericin B and nystatin) and 5-fluorocytosine (5-FC). The following mutation hot spots were identified: (1) ergosterol biosynthesis pathway mutations (ERG11 and UPC2), resulting in azole resistance; (2) overexpression of the efflux pumps, promoting azole resistance (transcription factor genes: *tac1* and *mrr1*; transporter genes: CDR1, CDR2, MDR1, PDR16 and SNQ2); (3) cell wall biosynthesis mutations (FKS1, FKS2 and PDR1), conferring resistance to echinocandins; (4) mutations of nucleic acid synthesis/repair genes (FCY1, FCY2 and FUR1), resulting in 5-FC resistance; and (5) biofilm production, promoting general antifungal resistance. This review also provides a summary of standardized inhibitory breakpoints obtained from international guidelines for prominent *Candida* species. Notably, *N. glabrata*, *P. kudriavzevii* and *C. auris* demonstrate fluconazole resistance.

## 1. Introduction 

### 1.1. Candidiasis

Candidiasis is an infection caused by the overgrowth of pathogenic yeast from the *Candida* genus [1]. Its prevalence accounts for the most common type of opportunistic fungal infection affecting human heath globally, with more than a billion cases on a yearly basis [2,3]. Typically, yeast can live harmlessly on the host’s mucosal tissues, such as in the oral cavity, gastrointestinal mucosa and vaginal mucosa, unless balance is disrupted [4,5]. The immunosuppressed, elderly population and palliative patients are highly susceptible to *Candida* infections [6]. Oral candidiasis results in local oral pain and discomfort, enhanced oral dryness, loss of taste and aversion to food and may lead to secondary complications [7,8]. Failure to treat candidemia in sufficient time is associated with a significant risk of mortality, especially in severe cases that have evolved into invasive fungal diseases (IFDs) [9,10].

Fungal resistance to traditional antifungal treatments has emerged as a significant and continued threat, yet it has received limited focus until recently in the fight against antimicrobial resistance [3,10]. Numerous factors contribute to the rising incidence and expanding geographic reach of pathogenic *Candida* infections. These includes the increase in immunocompromised patients, fungi continuing to evolve resistance to treatments and the limited access to timely diagnostic options for clinicians [10,11]. Furthermore, these fungal species grow more optimally at higher temperatures. As a result, global warming enhances the growing threat of fungal infection spread increasing beyond the load that health care can manage [11]. The worldwide impacts of pathogenic *Candida* and resistant infections include the increased burden on the healthcare system, higher costs and fatalities arising from treatment failures [10,12]. Knowledge of the molecular mechanisms underlying antifungal resistance is important to help drive the development of novel fungal therapeutics and diagnostics. Therefore, the overall objective of this literature review is to summarize the known molecular mechanisms associated with antifungal resistance in *Candida* infections. In particular, this review highlights the gene biomarkers and mutation profiles of antifungal resistance for the main antifungals currently available.

#### 1.1.1. Candida Species of Interest 

This review focuses on infectious *Candida* species that comprise most reported candidiasis cases, including *C. albicans*, *C. glabrata* (*Nakaseomyces glabrata*), *C. parapsilosis*, *C. tropicalis* and *P. kudriavzevii* (*Pichia kudriavzevii*) [13,14,15,16]. All five species listed here are among the 19 fungi included in the first fungal priority pathogens list (FPPL) recently released by the World Health Organization (WHO) [17]. In terms of priority ranking, *C. albicans* is critical; *P. kudriavzevii* is medium; and *N. glabrata*, *C. tropicalis* and *C. parapsilosis* are high [17].

Phylogenetic categorization of *Candida* yeast suggests polyphyly and pathogenic diversity among members [18]. Three of these species (*C. albicans*, *C. parapsilosis* and *C. tropicalis*) belong to the CTG clade, which contains most pathogenic *Candida* species, while *P. kudriavzevii* is more closely related to a wine-making yeast (*Brettanomyces bruxellensis)* [18]. Members of the CTG clade have a divergence in their genetic code compared to other Saccharomycotina subphylum yeast [19,20]. These species are categorized based on the CTG codon being transcribed and translated into serine instead of a typical leucine [19,20]. *C. glabrata* is part of the *Nakaseomyces* clade and was recently renamed *Nakaseomyces glabrata* for improved classification [21]. Despite this species being one of the few pathogenic members of its clade, it is the second most common cause of candidiasis globally [18]. The consequences of genetic alterations in *N. glabrata* may diverge from typical *Candida* species because it is a haploid organism [22]. Resistance could arise at a higher rate because a single recessive point mutation can present phenotypically due to a haploid genome. This contrasts with other *Candida* species that are diploid and therefore may require two copies of the mutated gene to present with resistance [23].

#### 1.1.2. *Candida auris*

*Candida auris* of the CTG clade is listed under critical priority in the FPPL due to its high infectiousness, global spread and high fatality risk [17,18,24,25]. Numerous species isolates have been identified as displaying resistance to several antifungals [26,27,28]. An update on *C. auris* released by Public Health Ontario (2023) indicated high rates of resistance to azole drug fluconazole (87–100%), while polyene amphotericin B and echinocandin resistances are cited less frequently, with ranges of 8–35% and 0–8%, respectively [29]. The CDC reported a similar rate of approximately 30% for polyene-resistant strains [30]. At least 4% of global cases of *C. auris* infections display multidrug resistance to all three antifungal types, which can make adequate clinical treatment especially difficult [29]. A detailed overview of the associated antifungal resistance mechanisms for *C. auris* highlights that similar genes are likely involved, as well as other related pathogenic members from the CTG clade [31]. Given the increased challenges in treating infections and, consequently, the spread of this highly pathogenic species, it is imperative to continue developing management strategies [10].

### 1.2. Primary/Intrinsic Resistance vs. Secondary/Acquired Resistance

Fungal resistance can be divided into primary/intrinsic and secondary/acquired resistance. Some fungal species are intrinsically resistant to a specific antifungal drug because of innate functional or structural attributes. This stable feature is seen in all strains from the same species and has not evolved due to previous antifungal exposure [1]. One example is the intrinsically fluconazole-resistant *P. kudriavzevii* [32,33]. Alternatively, acquired resistance can evolve in strains of a *Candida* species that are typically susceptible to an antifungals. This secondary form of resistance usually develops after prolonged treatment in a clinical or in vitro setting [34]. Mutations or chromosomal rearrangements can cause an overexpression of genes that override the effects of antifungal activity or the fungal stress response [34]. This change can revert to the original state once the pressure of the drug treatment is reduced or removed. Some mutants may retain the resistant phenotype regardless of future drug pressure [34].

Antifungal action can be evaded by pathogenic yeast by two main methods: (1) *An alteration of the interaction between drug and target*. This can result from either a change in the target protein amino acid (aa) sequence and, consequently, its structure or target protein overexpression. Alternatively, (2) *the cytoplasmic drug concentration can be reduced* via cell wall modifications that decrease drug absorption into the cell or the overexpression of efflux pumps that promote the export of drug molecules out of the cell [34].

Multidrug resistance (MDR) occurs when these mutations accumulate in the yeast genome in target pathways, which limits the amount of treatment options available [23,35]. An example of MDR was observed in *C. albicans* isolates with resistance to both fluconazole and clotrimazole [36]. Detecting mutant genotypes with acquired resistance in a timely manner could be an independent and useful predictive risk factor for treatment failure [34,37,38].

### 1.3. Standardized Measures of Susceptibility Testing 

Minimum inhibitory concentrations (MICs) calculated with in vitro broth microdilution susceptibility testing are used to categorize dose-dependent resistance in *Candida* species [39,40,41]. There are two main standardization methods that outline the established breakpoint concentrations for ranges of resistance: those of the ***Clinical Standards Laboratory Institute*** (**CLSI**—North America) and the ***European Committee on Antimicrobial Testing*** (**EUCAST)** [40,41,42]. The available CLSI and EUCAST breakpoint data for various antifungals, *Candida* and other related clinically relevant yeast species are summarized in Table 1. Considering that *C. parapsilosis* cryptic species *C. orthopsilosis* and *C. metapsilosis* are of low prevalence, the MIC breakpoint data (Table 1) for *C. parapsilosis* can be applied in cases when further species characterization has not been completed [43,44,45,46,47]. One trend identified in the data is the tendency of high-resistance concentrations in *N. glabrata* for fluconazole, whereas lower MICs are implicated with the use of echinocandins for this species.

Strains exposed to antifungal drugs can be described as susceptible (S) or clinically resistant (R). If the MIC is between the S and R cutoff values, then the intermediate (I) or susceptible dose-dependent (SDD) labels can be assigned depending on the standard used. Additionally, if breakpoint data are unavailable, epidemiological cutoff values (ECV) can provide guidance in distinguishing between a wild-type and resistant strain [46]. The ECV for an antifungal medication defines the upper limit of the drug concentration range that is typically sufficient to treat a wild-type member of a *Candida* species [46]. Strains exhibiting intermediate resistance can tolerate drug concentrations higher than typical MICs, which enables continued fungal growth. This feature is seen more often with fungistatic drugs and has been well characterized in *C. albicans* exposed to fluconazole [10].

### 1.4. Geographic Influence on Rates of Antifungal Resistance 

Geographical differences in resistance profiles have been observed for the various *Candida* species [48]. In terms of distribution, *C. albicans* and *N. glabrata* are the two most common species in the U.S., while *C. tropicalis* is most frequent in India [49]. The frequency of each antifungal prescribed to patients and the rates of resistance can also vary. In Iran, a meta-analysis identified resistance to at least one azole including clotrimazole (26%), ketoconazole (21%) and fluconazole (20%) among more than 5000 tested antifungal-resistant clinical isolates [50]. Rates of polyene resistance were also estimated for strains exposed to amphotericin B (7.3%) or Nystatin (4.4%), with echinocandin resistance evaluated for caspofungin (4.5%) and anidulafungin (1.8%) [50]. This coincides with a trend of higher rates of fluconazole resistance compared to echinocandins observed across various countries in the Ibero-America, Europe and Asia-Pacific regions [51,52,53,54]. The geographic variability in antifungal resistance profiles emphasizes the importance of the development of multinational surveillance registries for fungal infections (e.g., FungiScope™ CandiReg), as recommended by the WHO [17].

## 2. Antifungal Classes and Frequency of Resistance 

A range of antifungal drug classes is available to target various molecules and pathways associated with pathogenic *Candida* infections. The reviews by Bhattacharya et al. (2020) and Tilley and Tharmalingam (2022) provide an excellent summary of the four primary antifungal drug classes: azoles, polyenes, echinocandins and nucleoside analogs [1,55]. The section below summarizes the mechanisms of action for each antifungal drug class (Figure 1), as well as relevant resistance profiles. The molecular structures of drugs from each antifungal class and key points for each type are presented in Figure 2.

### 2.1. Azoles

Azoles are five-membered heterocyclic compounds classified into two groups based on the number of azole-ring nitrogen atoms: imidazoles with two nitrogens, like clotrimazole, ketoconazole and miconazole; and triazoles with three nitrogens, like fluconazole, itraconazole and voriconazole [55,56]. This antifungal class inhibits the production of ergosterol, an important component of the fungal cell membrane. With wide fungistatic activity, it is a cost-effective and relatively safe treatment option [57,58]. Fungistatic effects result in the inhibition of yeast growth [1]. Fluconazole has been prescribed as a first-line agent for fungal infections, and consequently, resistance has also been frequently cited [59]. The development of second-generation triazoles like voriconazole, posaconazole and isavuconazole offers secondary options for resistant *Candida* infections, although acquired resistance has been noted in past years [1]. Another barrier to the successful treatment of candidiasis infections with azoles is varying pharmacokinetics. Some drugs of this type, like itraconazole, may have poor absorption. For internal use, the absorption can be improved with food intake [60].

Different rates of azole resistance have been reported in clinical isolates depending on the species and antifungal. For *N. glabrata*, up to 10% of studied isolates were reported to be resistant to fluconazole [35,61]. Furthermore, resistant *N. glabrata* strains frequently display MDR or decreased susceptibility to other similar antifungals, including clotrimazole, itraconazole, posaconazole and voriconazole [62,63]. Azole cross resistance is also seen in isolates of other *Candida* species [63]. Overall, *N. glabrata* isolates have intrinsically higher MIC values for fluconazole compared to other related species [47,64]. The development of resistance in this species has been a concern for years [63,65]. Some cases of azole-resistant isolates may have higher virulence, which could subsequently increase resistant growth [66].

### 2.2. Polyenes

Polyene antifungals like amphotericin B and nystatin target the fungal plasma membrane by binding ergosterol molecules and forming pores that leak cell contents (monovalent ions K^+^, Na^+^, H^+^ and Cl^−^) [67]. These are potent agents with fungicidal and fungistatic activity that are used in clinical practice for their effectiveness despite relatively higher rates of toxic side effects like kidney/liver issues or anaphylaxis [50,68,69]. Fungicidal agents can kill infectious yeast cells directly [1]. To limit the possibility of treatment toxicity, this class is best used for topical infections such as in the oral cavity and for a limited time course [69]. Reports of isolated fungal strains with acquired polyene resistance are relatively rare, despite decades of use in the clinical setting [70,71]. This may be attributed to the effectiveness of their fungicidal activity in eliminating infections and thus preventing the evolution of stable resistant mutants. Additionally, decreased virulence is found in various *Candida* species that are polyene-resistant [66]. Amphotericin B may have limited effectiveness for *P. kudriavzevii* strains [72]. Notably, *C. auris* displays higher rates of antifungal resistance to amphotericin B than most related species [29,30]. Studies have linked this resistant phenotype with mutated genes involved in ergosterol biosynthesis, which results in an overall reduction in the drug target of ergosterol [31].

### 2.3. Echinocandins

Echinocandins including caspofungin, micafungin and anidulafungin target the fungal cell wall, a feature not found in mammalian cells [73]. Echinocandins are composed of cyclic hexapeptides with lipid side-chain modifications that enable antifungal action [74]. They inhibit the synthesis of a major cell wall component via non-competitive binding to the Fks1 subunit of the β1–3 glucan synthase enzyme [75]. This action promotes a fungicidal effect, as the cell wall integrity is compromised, with increased permeability and subsequent amino acid leakage [74].

This antifungal class was developed more recently in the 1990s and is typically effective against most *Candida* strains, including those displaying azole resistance [66,74]. Wiederhold (2017) recommended echinocandins as a good first-line treatment option for immunocompromised patients with recurring candidiasis infections and previous exposure to azole antifungals [48]. Garcia-Effron (2021) further supported use for initial antifungal treatment because no *Candida* species has been identified with intrinsic resistance [34]. Other recent studies highlight the high effectiveness of this class in the treatment of azole-resistant infections [50,76].

*C. albicans* tends to be the most susceptible to caspofungin, followed by *N. glabrata*, *C. tropicalis*, *P. kudriavzevii*, *C. parapsilosis* and *M. guilliermondii* [74]. The last two species listed seem to have more naturally arising FKS1 point mutations; thus, *C. parapsilosis* and *M. guilliermondii* appear to be more intrinsically echinocandin-resistant [77,78]. Specifically, the P660A (proline-to-alanine at amino acid position 660) intrinsic point mutation in the FKS1 gene is frequently found in isolates of the *Candida parapsilosis* family (*C. parapsilosis*, *C. orthopsilosis* and *C. metapsilosis*) [74].

Secondary resistance to echinocandins has been observed and linked to point mutations in the FKS1 gene that alter antifungal binding capacity [79]. Strain viability may be compromised as indicated by the reduced virulence seen in multiple echinocandin-resistant *Candida* species [66]. Resistant strains typically display this phenotype for all agents of this class [74]. Furthermore, these mutants usually do not show cross resistance to other antifungal treatments like amphotericin B or azoles [74]. In some cases, treatment results can be improved by switching to one or both of these two antifungal types [74].

### 2.4. 5FC

5-Fluorocytosine (5FC) can be used to target and disrupt nucleic acid biosynthesis within the cell [80]. This nucleoside analog used in conjunction with polyene amphotericin B is a reliable option for difficult-to-treat *Candida* infections and cryptococcal meningitis [81,82,83]. The minimum inhibitory concentration for 90% of fungal growth (MIC90) (National Committee for Clinical Laboratory Standards (NCCLS)) determined using the antifungal susceptibility testing method has been cited from 0.12 to 1 ug/mL depending on the species and sample [83]. Thus, it is an effective agent at relatively low doses for many key *Candida* species, including *C. albicans, N. glabrata* and *C. dubliniensis* [84]. However, *P. kudriavzevii* appears to have much higher intrinsic resistance, with the MIC90 threshold reached at 32 ug/mL and cells displaying limited sensitivity to 5FC [84].

## 3. The Ergosterol Biosynthesis Pathway and Antifungal Resistance

Azoles and polyenes target the ergosterol biosynthesis pathway, specifically the 14α–demethylase enzyme (Erg11) and ergosterol molecules respectively (Figure 3) [1]. Sterols, along with sphingolipids, can form lipid rafts within the fungal cell membrane that contain proteins crucial for yeast survival, like stress response, signaling and nutrient transport proteins [1]. There are 25 different pathway enzymes involved in the formation of ergosterol [85]. Azoles primarily exhibit fungistatic action via non-competitive binding to the Erg11 enzymatic active site, which inhibits its activity and results in an overall decrease in cellular levels of ergosterol [86].

### 3.1. ERG11

By 2010, over 160 amino acid substitutions had been identified in the ERG11 gene, each with varying genetic consequences [87,88,89]. Many single substitutions are synonymous and have no impact on gene function (Table 2). In addition, non-synonymous single-nucleotide changes occurring in *Candida* strains do not inherently contribute to antifungal resistance (Table 2). For example, White et al. (2002) identified D116E and E266D, the two most frequent ERG11 substitutions in one sample set, using restriction fragment length polymorphism (RFLP) analysis, with no consistent correlation [36]. These instances indicate that there is natural genetic variation within each fungal species and that allelic polymorphisms are relatively common. Mutations present in both resistant and susceptible samples are an indicator that the alteration is not directly implicated in conferring antifungal resistance [86].

*Candida* spp. have acquired azole resistance via ERG11 point mutations that typically lower azole binding affinity to the Erg11 active site. Additionally, gain-of-function mutations in upstream transcriptional regulators can increase ERG11 expression and confer resistance. [34,86]. Point mutations resulting in a defective Erg11 enzyme unable to bind to azoles have been clustered in three hotspot regions: 105–165, 266–287 and 405–488 [94]. Xiang et al. (2013) studied clinical isolates and reported numerous single substitutions in ERG11 that conferred fluconazole resistance; a subset also resulted in voriconazole-resistant strains [86,90]. Some missense polymorphisms, like the fluconazole-resistant Y132F mutation, have been identified in multiple different species, including *C. albicans*, *N. glabrata* and *C. tropicalis* [94,97,102]. In addition, four ERG11 substitutions conferring fluconazole resistance were identified in an in vitro experimental setting [91,92]. The ERG11 mutations identified in different species are listed in Table 2 [28,95,96,98].

Polymorphisms can have different biological impacts, depending on whether they are singly present or in combination with other relevant SNPs. Resistance levels can be enhanced when some mutations with moderately low impact alone are present simultaneously with other resistance mutations [86,103,104]. One example is increased FLZ, ketoconazole and ITZ resistance observed when the Y132H polymorphism was present in combination with S405F or R467K [104,105]. Alternatively, the S405K mutation showed some resistant effects alone, but in conjunction with other SNPs, the samples with this mutation were susceptible [106].

The variety of currently available azole medications have different structural features (e.g., short and long chains); therefore, mutations affecting their efficacy may differ. For example, ERG11 point mutations K128T and Y132H may affect the ability of fluconazole or voriconazole molecules to enter or bind the target active site. Mutations in other gene sequences can also confer resistance, such as the G464S mutation, which affects haem coordination due to its location near a key cysteine residue [93]. These mutations do not have the same binding inefficiency for posaconazole and itraconazole treatments, suggesting that these two antifungals have other key interaction sites within the Erg11 protein [93]. Indeed, the long chains added to the posaconazole and itraconazole molecules may provide the additional contact points needed to stabilize drug–protein binding despite the presence of affecting mutations [106].

### 3.2. Mutations in Transcriptional Regulators

Zn2-Cys6 transcription factor *uptake control 2* (Upc2) regulates most of the genes in the ergosterol biosynthesis pathway on some level (Figure 3). Gene overexpression of UPC2 can be induced upon azole exposure and can sufficiently compensate for the inhibition of target enzymes. Gain-of-function (GOF) mutations in UPC2 can drive this gene overexpression and fluconazole resistance [101]. For example, a series of studies of azole-resistant clinical isolates identified the A643V substitution in the UPC2 gene, which was validated in vitro to confer resistance [101,107]. This mutation and possibly others within this gene sequence region (G648D) may cause Upc2 to be released from a repressor, inducing hyperactivity [107]. However, alternate models exist that involve the sterol regulator, SREBP [107]. Regardless, the UPC2 A643V mutation affects the C-terminal regulatory domain and, consequently, normal UPC2 function [107].

Other UPC2 GOF mutations reported from clinical isolates to exhibit azole resistance are listed in Table 2 [100,101,108,109,110]. A genome-wide ChIP (chromatin immunoprecipitation) study used to identify Upc2-bound gene promoters identified up to 202 genes, including UPC2 itself [111]. Other upregulated genes were found to be involved in ergosterol biosynthesis; oxidoreductase activity; and numerous drug efflux pumps, including MDR1 (MFS-transporter) and CDR1 (ABC-transporter) [101]. Considering that the overexpression of both Erg and efflux pump genes is implicated in antifungal resistance, UPC2 is a good target for the detection of mutations that predict antifungal resistance in clinical patients.

### 3.3. Other ERG Genes and Toxic Diol Formation

The inhibition of Erg11 alters pathway products and induces the synthesis of a fungistatic toxic diol (14α-methylergosta 8–24 (28) dienol) by downstream enzymes (Erg3, Erg6, Erg25, Erg26 and Erg27) [1]. Erg3 is a C5 sterol desaturase enzyme needed for the conversion of episterol to ergostatrienol [112]. When its expression is inhibited by mutation or deletion, the reduction in toxicity due to the inhibition of toxic diol formation is sufficient to confer resistance in some *Candida* spp. [113,114,115]. However, ERG3 inactivation appeared to minimally contribute to azole resistance in a wide range of studied clinical *C. albicans* [116]. Q139A substitution in ERG3 has been identified from *N. glabrata* clinical isolates with azole resistance (Table 2) [99].

Deletion of ERG pathway genes such as the ERG6 gene can impact resistance to other antifungals. This ∆24 sterol C-methyl transferase is non-essential for ergosterol biosynthesis but it is needed for toxic diol formation. Its disruption contributes to azole resistance in *C. albicans* [117,118]. Furthermore, resistance to the polyene amphotericin B has been cited due to loss-of-function alterations in ERG6 and other Erg genes, including ERG2, ERG3, ERG5 and ERG11 [66]. Still, the relative infrequency of amphotericin B-resistant *Candida* clinical strains suggests that these ERG gene mutations come at a cost of fitness or pathogenicity [57,119].

One differential gene expression analysis of a lab-generated *C. albicans* strain with resistance to fluconazole and amphotericin B identified numerous upregulated ergosterol pathway genes, including ERG5, ERG6 and ERG25 [112]. Additionally, genes involved in cell stress responses were found to be upregulated, including DDR48 and RTA2 [112]. In *C. albicans*, RTA2 is a key gene in calcium signaling pathways, and it has been shown to modulate azole resistance, including biofilms [120]. There is no mammalian RTA2 homolog gene, so this may be a prime target for future development of more effective antifungals [120]. The overexpression of these select ERG genes may alter the biosynthesis pathway products at key points and consequently reduce antifungal susceptibility [112].

## 4. Cell Membrane Proteins and Antifungal Resistance

Two types of membrane transporters have been implicated in azole resistance: ABC-Ts (ATP-binding cassette transporters) and MFS-Transporters (major facilitator superfamily transporters) [36,110,121,122]. ABC-Ts facilitate the movement of molecules across membranes using energy derived from ATP hydrolysis, while MFS-Ts require a proton gradient across the plasma membrane to transport foreign molecules out of the cell [1]. Both types of transporters can bind azoles as a substrate, and the drug can be exported out of the cell. This decreases the intracellular drug concentration and allows cells to circumvent the antifungal effects [1]. An additional MLT1 (ABC-T) transporter has been implicated in *C. albicans* resistance. This multidrug resistance protein (MRP) is localized to the vacuolar membrane and can import azole molecules into the vacuole for sequestration. Mutations in the MLT1 sequence can cause incorrect localization or the inability to bind and transport azoles (Table 3) [123].

### 4.1. Drug Efflux Pump/Transporter Genes and Resistant Mutations 

Despite the large number of transporters found in the *C. albicans* genome, evidence of transporter overexpression in resistant clinical isolates is currently limited to ABC-Ts CDR1 and CDR2 and MFS-Ts MDR1 and PDR16 (Figure 3) [1,111,131,133]. CDR1 and CDR2 overexpression is frequently observed in clinical isolates, and coregulation of these two pumps is evident [36]. In fact, prolonged exposure to azoles can result in trisomy development of chromosome 3, which encodes CDR1 and CDR2, as well as subsequent overexpression of these genes [124]. For azole-resistant *N. glabrata* clinical isolates, overexpression of ABC-T genes CDR1, PDH1 (CDR2 in *C. albicans*) and SNQ2 and MFS-T genes QDR2, FLR1 and PDR16 has been cited [62,104,134,135,136,137,138,139]. In some cases, SNQ2 overexpression alone was enough to produce azole-resistant phenotypes in *N. glabrata* isolates [62]. Other MFS-Ts that may be implicated in antifungal species for *C. albicans* and *N. glabrata* are FLU1 and TPO3, respectively [136,140]. Similar drug transporters ABC1 and ABC2 in *P. kudriavzevii* were implicated in antifungal resistance but likely with a supplementary role [121]. Azole-resistant *C. parapsilosis* and *C. dubliniensis* strains with increased drug efflux have also been identified [141,142]. Considering that multiple studies cite efflux pump overexpression alone is sufficient to confer azole resistance, the detection of overexpressed pumps may serve as a biomarker of antifungal resistance [36,62,143].

### 4.2. Transcriptional Regulators of Transporter Genes 

Efflux pump overexpression can be further induced by upstream GOF mutations, regulating transcription factor genes (Table 3) [59,125,126,127,128,129,130,144,145]. In *C. albicans,* the CDR1, CDR2 and PDR16 transporters, as well as MDR1, are regulated by zinc-cluster (Zn2-Cys6) transcription factors Tac1 and Mrr1, respectively [111,121,131,133,144,146]. Other potential transcriptional regulators of CDR1 expression include Tup1 (thymidine uptake 1) and Ncb2 (β subunit of the NC2 complex) [147]. For MDR1, additional transcriptional regulators are Cap1 (bZIP transcription factor) and Mcm1, but no antifungal-resistant mutations have been identified to date [148,149]. Additionally, Dunkel, Liu et al. (2008) found that fluconazole-resistant *C. albicans* strains with the G648D mutation in UPC2 exhibited MDR1 upregulation [101].

Efflux pumps CDR1, SNQ2, PDH1 and QDR2 implicated in *N. glabrata* antifungal resistance are regulated by the Pdr1 transcription factor [132,137,150]. Mutations in the PDR1 sequence can confer gene overexpression and result in the upregulation of numerous downstream targets (Table 3) [116,151,152]. GOF mutations within the PDR1 sequence regions linked to azole resistance corresponded to putative inhibitory (aa 312–382) and transcriptional activation (aa 800–1107) domains, as well as aa 539–632, coding for the middle homology region [132,152]. The PDR1 gene may be particularly susceptible to hypermutability due to the coinciding high mutation frequency seen in msh2 (mismatch repair gene 2) [23].

### 4.3. Post-Translational Regulation of Transporter Genes

In addition to transcriptional and translational control of efflux pump genes, there is evidence of post-translational regulation. For instance, mitochondrial biogenesis gene FZO1 is important for directing Cdr1 to the correct membrane [153]. In FZO1 deletion mutants, Cdr1 was found to be mis-sorted to the vacuole, which was correlated with increased azole susceptibility [153]. Other examples are poly(A) polymerase 1 homozygosity and hyperadenylation, as observed in azole-resistant clinical isolates, which correlated with increased CDR1 mRNA stability [154].

## 5. The Cell Wall Biosynthesis Pathway and Antifungal Resistance

The fungal cell wall is a structural feature of pathogenic yeast that is absent in human cells, making it a good target for antifungals. The cell wall in fungi is mostly composed of β1–3-glucan and chitin polysaccharides that are covalently cross-linked to form carbohydrate polymers [155]. Other sections include inner cell wall proteins linked to mannose and galactose polysaccharides [74]. The overall cell wall architecture comprising these various molecules is consistently monitored to maintain cell viability [74]. Defects or the removal of any key aspect of the cell wall structure typically result in lethality [74].

### 5.1. FKS1 and FKS2 Sequence Mutations

Echinocandin resistance is associated with modifications to the FKS1 or FKS2 gene sequences, which code for the β1–3 glucan synthase enzyme (Figure 3) [156]. FKS1 mutations have been identified in resistant *C. albicans*, *C. tropicalis*, *P. kudriavzevii* and *N. glabrata*, while FKS2 mutations have only been identified in *N. glabrata* [157,158,159,160]. No definitive intrinsic resistance has been established in any *Candida* species, but secondary resistance can be acquired in individual isolates through point mutations. Notably, *C. parapsilosis* and *Meyerozyma guilliermondii* have a higher rate of spontaneously occurring FKS1 point mutations and may be considered more intrinsically resistant [77,78]. For example, point mutation P660A in the FKS1 gene is believed to confer some intrinsic reduction in caspofungin susceptibility and is found in all *C. parapsilosis* family members [161].

Significant GOF mutations in the FKS1 gene sequence in clinical isolates have been described, particularly in the hotspot regions of 637–654 and 1345–1365 (Table 4) [74,157,162,163,164,165,166,167,168]. Walker et al. (2010) noted that non-synonymous substitutions at aa position 645 have been commonly observed. Here, serine substitutions with phenylalanine, proline or tyrosine have been cited [163,164,169]. Hotspot mutations are usually dominant, and *C. albicans* fungal cells only require one mutant allele for resistance to be conferred across the three echinocandins [74]. However, in vitro experiments suggest that there is a fitness disadvantage for FKS1 mutant *C. albicans* strains, which may limit population spread for these mutants under non-echinocandin treatment conditions [170]. This is consistent with the generally low prevalence of FKS1 mutations described in the literature for various *Candida* species [170].

Evidence of FKS2-resistant mutations were previously limited to in vitro experiments, but recently, such mutations have been identified in clinical *N. glabrata* isolates (Table 4) [34,35,173]. This coincides with evidence suggesting that FKS2 has higher levels of expression in *N. glabrata* than FKS1 and that mutations may have a greater influence on echinocandin resistance in this species [158]. Furthermore, a study found a relatively high natural mutation frequency in *N. glabrata* cells under echinocandin drug pressure for both FKS1 and FKS2, with twice the rate identified for FKS2 [171,172].

Bienvenu et al. (2019) highlighted the *N. glabrata* S629P (FKS1) and S663P (FKS2) mutations and indicated that genotyping of these regions is an accurate indicator of resistance to echinocandins [48]. A relatively small number of mutations in FKS1 seemingly accounts for most of the echinocandin-resistant *Candida* strains. As a result, these validated gain-of-function mutations are one example of a potential biomarker for antifungal resistance. PCR assays were developed to detect these mutations, which has aided in treatment, although there is still a lack of a timeliness in achieving this result [10,174]. Recent CLSI standards state that caspofungin resistance conferred by hotspot FKS1 mutations is best validated by testing with an additional echinocandin (e.g., anidulafungin or micafungin) or DNA sequencing analysis of the relevant genomic region [175,176]. Overall, the development of a rapid clinical test for the identification of FKS1 mutations may aid in efficient diagnosis and prescriptions at the time of identification of the fungal infection [10].

### 5.2. Transcriptional Regulators of fks Genes

Transcriptional regulators upstream of *fks* genes can also affect echinocandin resistance. In particular, point mutations in transcription factor PDR1 have been found in numerous resistant *N. glabrata* isolates (Figure 3). Transcription factor Pdr1 is detailed further in Section 4. In addition, there is evidence that in *N. glabrata*, the Upc2 transcription factor detailed in Section 3 is involved in FKS1 coregulation [177].

### 5.3. Protein Analysis Associated with Echinocandin Resistance

Cell wall remodeling enzymes upregulated in previous protein studies include glucanosyl transferases Phr1, Phr2 and Crh, as well as chitin-glucanosyl transferase family proteins [178,179]. Proteomic analysis conducted using mass spectrometry (LC-MS/MS) revealed that levels of cell wall organization and maintenance proteins can differ between drug-resistant and susceptible strains in response to caspofungin treatment [180,181]. Differentially expressed enzymes related to cell wall synthesis and remodeling include Sun41, Gsc1, Pmt1, Mnt1, Als3, Als4, Ecm33 and Pga31 [181]. Validated caspofungin tolerance regulators Cas5, Mkc1, Swi4, Gin4, Stt4, Ahr1 and Pkc1 were detected in an alternate screen [182]. Furthermore, metabolic enzymes with immunogenic activity including Eno1, Fba1, Gpm1 and Pgk1 have also been observed to be released in *Candida* cells exposed to caspofungin [181,183]. Finally, the Hsp90 molecule may have a regulatory role with key resistance regulators like Mkc1 from the Pkc1 signaling pathway [184,185]. These protein subsets may be good *Candidates* for diagnostic markers that predict echinocandin resistance in *Candida* species. Further validation across a wide range of antifungal-resistant clinical isolates is still needed [181].

Enzymes in cell wall salvage pathways are additional targets for echinocandin resistance biomarkers. Fungal cells can compensate for and strengthen the cell wall via an upregulation of chitin synthesis genes in response to cell wall damage induced by antifungal treatment. This mechanism has been observed in *C. albicans*, while *N. glabrata* and *P. kudriavzevii* isolates showed no such increase in chitin content or resistant growth. [74,186]. The upregulation of chitin synthesis involves the induction of PKC (protein kinase C), calcium/calcineurin and HOG (high-osmolarity glycerol response-MAP-K activated) signaling pathways [187]. *C. albicans* cells exposed to activators of these pathways were found to have higher chitin contents and decreased caspofungin susceptibility [74,188].

## 6. The Nucleic Acid Biosynthesis Pathway and Antifungal Resistance 

The biosynthesis of nucleic acids (DNA and RNA) and subsequent protein synthesis in pathogenic fungi can be targeted with nucleoside analogue 5-fluorocytosine (5-FC). As a prodrug, it requires activation within the fungal cell via metabolism by the pyrimidine salvage pathway [83]. Then, it is incorporated as a toxic substrate, and the affected nucleotides have damaging effects on cell viability [83]. Membrane permeases encoded by FCY2 (cytosine permease) and other homologs (FCY21 and FCY22) are responsible for the active transport of 5-FC into the cell (Figure 3) [83]. 5-FC is then converted to toxic 5-fluoro-uridylate by enzymes encoded by *fcy1* (cytosine deaminase) and FUR1 (uracil phosphoribosyltransferase (UPRT)) [83]. The FCY1 homologue in *C. albicans* and other *Candida* species is the FCA1 gene [189,190]. The lack of cytosine deaminase in mammalian cells prevents 5-FC conversion and subsequent toxic effects [191].

Resistance to 5-FC could arise with mutation or loss of any of the three key enzymes (FCY1, FCY2 or FUR1), as discovered in model organism yeast *Saccharomyces* cerevisiae [192,193]. Increased pyrimidine production in the fungal cell can also serve to circumvent toxic antifungal activity [82,83]. Kern et al. (1991) were among the first to identify the correlation between a point mutation (Arg134Ser) in the FUR1 gene and 5-FC resistance in *S. cerevisiae* yeast cells [194]. The non-synonymous mutations in the FUR1, FCY1/FCA1 and FCY2 genes described in 5-FC-resistant clinical *Candida* samples are presented in Table 5 [195,196,197,198,199,200,201,202].

## 7. Biofilm Formation and Antifungal Resistance 

Yeast cells can grow freely/planktonically or develop an extracellular matrix (ECM) to attach to and form a highly organized community of microbial cells [203,204]. The ECM is composed of polysaccharides and proteins and can be produced by yeast at the site of infection to protect from antifungals and other stressors, including the host’s immune system [205,206,207]. As a result, *Candida* biofilms have intrinsic antifungal resistance [208]. The architecture of the biofilms is carefully structured to provide adequate space for nutrients and waste to pass in and out, respectively [209,210,211]. *C. albicans* is most often associated with biofilm formation [212,213,214]. Other *Candida* species that can form biofilms include *C. auris*, *N. glabrata*, *C. dubliniensis*, *P. kudriavzevii*, *C. parapsilosis* and *C. tropicalis* [203,215,216].

### 7.1. Biofilm Formation during Antifungal Treatment 

Over the course of antifungal treatment, a biofilm can develop and enable yeast cells to become more resistant [209,215]. Numerous classes of antifungals, including polyenes and azoles, have been cited as less effective over time, even within 72 h of biofilm development/maturation [209]. For fluconazole treatment, resistance in *C. albicans* with biofilms can be increased by up to 1000-fold compared to planktonic cells [215,217]. Alternatively, an in vitro study of *C. albicans* biofilms compared to planktonic cells demonstrated an approximately 10-fold increase in amphotericin B resistance [218]. One study found that caspofungin was effective for susceptible isolates, but limited efficacy was observed for resistant samples with biofilm production [181].

Multiple features of biofilms enable antifungal resistance aside from genetic mutations that typically drive resistance in planktonic cells, which adds to the complexity of treating this type of infection [219]. These include physical protection, increased cell density of the microbe community, persister cells and extracellular vesicular secretion [208,219,220]. Concurrently, the concentrations of drug doses required to inhibit the infection can quickly exceed what is clinically safe and available [215,221]. It is important to consider if this feature is present at the time of diagnosis or develops thereafter when detecting resistance in pathogenic yeast. The increased expression of the ALS3 gene encoding for a glycoprotein on the cell surface has been implicated as a potential biomarker of biofilm formation [222].

### 7.2. The Roles of β-1,3 Glucan and Biofilm-Associated Antifungal Resistance 

One of the main ECM components is β-1,3 glucan, which is synthesized by echinocandin target gene FKS1 [223]. The polysaccharide molecules are primarily responsible for the sequestration of antifungal molecules, making them a prime target for treatment of *Candida* infections with biofilms [224]. Supplemental treatment with the β-1,3 glucanase enzyme can break down this molecule and subsequently disrupt the biofilm architecture [225,226]. Furthermore, overexpression of the FKS1 gene can increase β-1,3 glucan production in *Candida* biofilms, and multidrug resistance has been cited, including against azoles, echinocandins and polyenes [227]. Other targets that may address the issue of drug sequestration by beta-glucans are glucan transferases such as Bgl2 and Phr1, as well as exoglucanase Xog1 [228,229]. These three enzymes can modify glucan and are involved in its post-translational transport from the cell to the ECM [208,229].

### 7.3. Relevant Antifungal Resistance Genes in Biofilm-Associated Candida Infections

*Candida* strains associated with biofilm formation and resistance to antifungal treatment may have similar key genes implicated in planktonic resistant isolates. Changes in the expression of ergosterol biosynthesis pathway genes and in biofilm membrane composition appear to be linked to subsequent azole and polyene resistance [230,231,232,233]. Differential gene expression of beta-glucan synthesis-associated genes SKN1 and KRE1 was observed in biofilm-associated-resistant *Candida* exposed to amphotericin B, in agreement with previous studies [218,230]. These relevant genes may be highlighted in the search for a suitable resistance biomarker. As a preventative treatment, farnesol can target ERG and MDR1 gene expression and downregulate these genes prior to the start of biofilm formation and fluconazole treatment [234].

Drug efflux pumps CDR1, CDR2 and MDR1, which are associated with azole resistance, may also be upregulated in biofilm-associated *Candida* strains [235,236,237,238,239,240]. Interestingly, neither polyenes nor echinocandins have been implicated as a substrate for drug efflux pumps in *Candida* infections of this type. This suggests that echinocandins could be a preferable treatment choice over azoles [236,237,241]. In fact, the use of echinocandins in combination with other antifungals may be one of the more potent options for treating a biofilm-associated infection, as seen in a study testing pharmaceutical combinations of echinocandin and liposomal amphotericin B (AmBisome) [242,243]. Improving the delivery system for antifungals to better access yeast cells using lipid vesicles can reduce the effects of resistance caused by biofilms [244]. Therefore, the development of nanoparticle delivery systems could be possible for other compounds being investigated for synergistic effects with traditional antifungals, as seen in a cinnamaldehyde study [245]. A detailed review of the range of natural compounds under investigation for treatment of *Candida* infections and biofilms is provided in [246].

## 8. Future Directions

With antifungal resistance being a continued problem, there is a need for the development of quick and reliable molecular diagnostic tests that detect organisms with intrinsic and/or secondary resistance due to the genetic mechanisms presented in this review [34]. Additionally, faster methods of species identification would be useful, given the differences in frequency and impact of antifungal resistance among *Candida* species. Currently, PCR-based methods using fungal cultures are still the first option for species identification and detection of antifungal resistance in individual strains [105,167]. The usefulness of real-time testing for resistance in *Candida* species to modify the treatment course has been well documented. For example, two patient cases of candidemia presenting with fluconazole-resistant *C. albicans* strains were successfully treated with amphotericin B, despite higher cost and risk of greater side effects [247].

Resistance can be acquired through a dynamic combination of numerous point mutations and other genetic or transcriptional alterations. A large-scale comparison between matched fluconazole-resistant and -susceptible *C. albicans* clinical isolates using microarray analysis identified almost two hundred genes (*n* = 198) that were differentially expressed [248]. In resistant isolates, multidrug resistance and oxidative stress response genes, among others, were found to be upregulated compared to susceptible samples [248]. This highlights the fact that a dynamic response that leads to antifungal resistance and identification of reliable biomarkers or gene expression profiles that different strains have in common would be beneficial to improve future treatment decisions.

To develop a dependable point-of-care (POCT) diagnostic assay for antifungal resistance, reliable biomarkers in *Candida* species need to be established. This review presents numerous possible gene mutation biomarkers that have been associated with antifungal resistance in clinical isolates from studies worldwide. It is important to note that different non-synonymous substitutions and other genetic alterations may result in similar genetic effects. In this scenario, assessing the mRNA overexpression of ERG11 may be a good biomarker for antifungal resistance, especially azoles. Pfaller et al. (2006) suggested that a test that can identify resistant *Candida* strains with high MICs would be more clinically useful than predicting susceptibility according to low MICs [63].

The emergence of new technologies, including next-generation sequencing, CRISPR and isothermal amplification-based detection assays, has enabled progress in the development of reliable assays to detect pathogenic nucleic acid profiles [249,250,251,252]. A review by Garcia-Effron (2020) provides a good summary of the available commercial kits and in-house methods in use for the detection of intrinsic and acquired antifungal resistance [34]. A novel antifungal POCT assay conducted using one of these methods would allow for quick diagnosis of potential antifungal resistance prior to initiating treatment. The yeast strain isolates obtained from a patient’s initial diagnostic healthcare visit could be tested to detect resistance biomarkers in their genetic profile. Then, clinicians could choose to avoid the first-line antifungal options in favor of other agents that target an alternate pathway or a next-generation antifungal agent if needed. For example, VT-1129, VT-1161 and VT-1598 are all Cyp51 (fungal lanosterol 14a-demethylase)-specific inhibitors with modifications that improve antifungal action [48]. Continued efforts to genetically and molecularly characterize a variety of antifungal-resistant *Candida* clinical isolate samples could elucidate other drug targets and support the development of alternative antifungal treatments [94].

Also, the development of antifungal vaccines would be aided by adequate characterization of pathogenic *Candida.* The identification of fungal surface proteins with immunogenic properties and, ideally, no additional side effects in humans could be used to produce an mRNA vaccine. The proactive immune response elicited when these proteins are expressed within the human body could protect the individual in defending against candidiasis infections in the future. The recent progress in mRNA vaccine technology for viruses has made this an important and promising option to address fungal infections [11].

Furthermore, this review of antifungal resistance highlights the diversity of members of the *Candida* genus and their differing responses to antifungal treatments. Notably, *N. glabrata*, *C. krusei* and *C. auris* are all highly fluconazole-resistant. The ability to identify these fungal species at the time of candidiasis diagnosis could improve treatment selection by prescribing echinocandins, polyenes or some combination thereof where appropriate. Overall, understanding the mechanisms of antifungal resistance in various pathogenic *Candida* can aid in developing POCT assays to identify resistant strains and positively contribute to clinical outcomes in the management of candidiasis.

## Figures and Tables

**Figure 1 cells-12-02655-f001:**
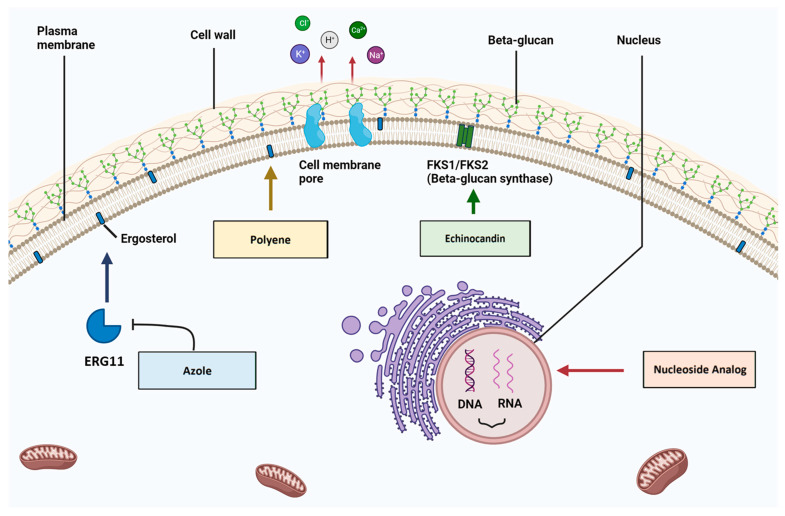
Mechanisms of antifungal action for the four main drug types. (1) **Azoles** bind to and inhibit the Erg11 enzyme and subsequent ergosterol production. (2) **Polyenes** bind to ergosterol and induce the formation of cell membrane pores, which cause intracellular ion leakage. (3) **Echinocandins** bind to and inhibit beta-glucan synthase, which disrupts cell wall architecture. (4) **Nucleoside analogues** are incorporated into nucleic acid molecules and disrupt DNA/RNA biosynthesis (created with BioRender.com, accessed on 16 October 2023).

**Figure 2 cells-12-02655-f002:**
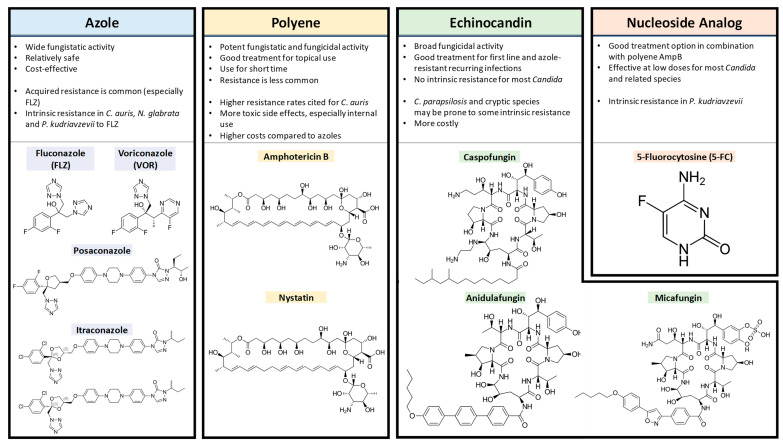
The key points for each of the four antifungal drug types with the chemical structures of members from each class. All drug structure images were obtained from *Wikimedia Commons*.

**Figure 3 cells-12-02655-f003:**
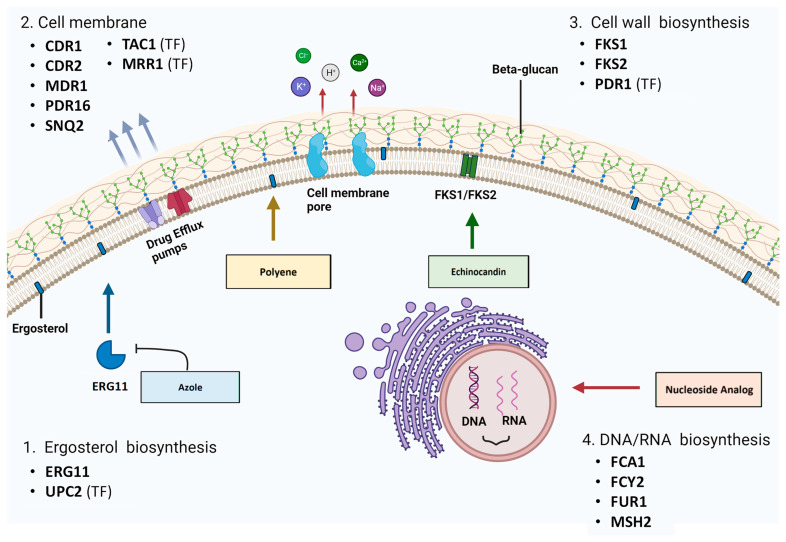
Genes associated with antifungal resistance in drug target pathways: (1) ergosterol biosynthesis, (2) cell membrane, (3) cell wall biosynthesis and (4) DNA/RNA biosynthesis (created with BioRender.com, accessed on 16 October 2023).

**Table 1 cells-12-02655-t001:** Antifungal breakpoint concentrations for various *Candida* and other related clinically important yeast species available through CLSI or EUCAST standards [32,40,41,42]. Abbreviations: S, sensitive; I, intermediate; SDD, susceptible dose-dependent; R, resistant; MIC, minimum inhibitory concentration.

Antifungal Class	Drug Name	Fungal Species	CLSI MIC Breakpoints (µg/mL)				EUCAST MIC Breakpoints (µg/mL)			
			S	I	SSD	R	S	I	SSD	R
**Azole**	**Fluconazole (FLZ)**	*C. albicans*	≤2	-	4	≥8	≤2	4	-	>4
		*C. dubliniensis*	-	-	-	-	≤2	4	-	>4
		*N. glabrata*	-	-	≤32	≥64	≤0.001	≤16	-	>16
		*P. kudriavzevii*	-	-	-	-	-	-	-	-
		*C. parapsilosis*	≤2		4	≥8	≤2	4	-	>4
		*C. tropicalis*	≤2		4	≥8	≤2	4	-	>4
	**Voriconazole (VOR)**	*C. albicans*	≤0.12	0.25–0.5	-	≥1	≤0.06	0.125–0.25	-	>0.25
		*C. dubliniensis*					≤0.06	0.125–0.25	-	>0.25
		*N. glabrata*	-	-	-	-	-	-	-	-
		*P. kudriavzevii*	≤0.5	1	-	≥2	-	-	-	-
		*C. parapsilosis*	≤0.12	0.25–0.5	-	≥1	≤0.125	0.25	-	>0.25
		*C. tropicalis*	≤0.12	0.25–0.5	-	≥1	≤0.125	0.25	-	>0.25
	**Posaconazole**	*C. albicans*	-	-	-	-	≤0.06	-	-	>0.06
		*C. dubliniensis*	-	-	-	-	≤0.06	-	-	>0.06
		*C. parapsilosis*	-	-	-	-	≤0.06	-	-	>0.06
		*C. tropicalis*	-	-	-	-	≤0.06	-	-	>0.06
	**Itraconazole**	*C. albicans*	-	-	-	-	≤0.06	-	-	>0.06
		*C. dubliniensis*	-	-	-	-	≤0.06	-	-	>0.06
		*C. parapsilosis*	-	-	-	-	≤0.125	-	-	>0.125
		*C. tropicalis*	-	-	-	-	≤0.125	-	-	>0.125
**Echinocandin**	**Caspofungin**	*C. albicans*	≤0.25	0.5	-	≥1	-	-	-	-
		*N. glabrata*	≤0.12	0.25	-	≥0.5	-	-	-	-
		*M. guilliermondii*	≤2	4	-	≥8	-	-	-	-
		*P. kudriavzevii*	≤0.25	0.5	-	≥1	-	-	-	-
		*C. parapsilosis*	≤2	4	-	≥8	-	-	-	-
		*C. tropicalis*	≤0.25	0.5	-	≥1	-	-	-	-
	**Anidulafungin**	*C. albicans*	≤0.25	0.5	-	≥1	≤0.03	-	-	>0.03
		*N. glabrata*	≤0.12	0.25		≥0.5	≤0.06	-	-	>0.06
		*M. guilliermondii*	≤2	4		≥8	-	-	-	-
		*P. kudriavzevii*	≤0.25	0.5		≥1	≤0.06	-	-	>0.06
		*C. parapsilosis*	≤2	4		≥8	≤4	-	-	>4
		*C. tropicalis*	≤0.25	0.5		≥1	≤0.06	-	-	>0.06
	**Micafungin**	*C. albicans*	≤0.25	0.5	-	≥1	≤0.016	-	-	>0.016
		*N. glabrata*	≤0.06	0.12	-	≥0.25	≤0.03	-	-	>0.03
		*M. guilliermondii*	≤2	4	-	≥8	-	-	-	-
		*P. kudriavzevii*	≤0.25	0.5	-	≥1	-	-	-	-
		*C. parapsilosis*	≤2	4	-	≥8	≤2	-	-	>2
		*C. tropicalis*	≤0.25	0.5	-	≥1	-	-	-	-
**Polyene**	**Amphotericin B**	*C. albicans*					≤1	-	-	>1
		*C. dubliniensis*					≤1	-	-	>1
		*N. glabrata*					≤1	-	-	>1
		*P. kudriavzevii*					≤1	-	-	>1
		*C. parapsilosis*					≤1	-	-	>1
		*C.tropicalis*					≤1	-	-	>1
		*C. auris*	Tentative breakpoints based on a mouse model reported by the CDC (2020): **S (≤1)**, **R (≥2)**							
	**Nystatin**	*Candida*	CLSI and EUCAST MIC breakpoints unavailable.Broth microdilution estimates based on *Brito* et al., 2011: **S (≤4)**, **I (8–32)**, **R (≥64)**							

**Table 2 cells-12-02655-t002:** Mutations in ergosterol biosynthesis pathway genes in pathogenic *Candida* species implicated in antifungal resistance.

Gene	*Candida* Species	Mutation	Type of Mutation	Antifungal Resistance	Location	Isolate Type	Ref.
**ERG11** (lanosterol 14a-demethylase)	*C. albicans*	Hotspot regions: aa105–165, 266–287 and 405–488	Substitution	Azole	USA	Clinical	[36]
		A61V, S405F, G448E, F449S, G464S, R467K and I471T	Non-synonymous substitution	Fluconazole	China	Clinical	[86]
		Y132H, Y132F, K143R and K143Q	Non-synonymous substitution	Fluconazole and voriconazole	China	Clinical	[86,89]
		A114S and Y257H	Non-synonymous substitution	Fluconazole and voriconazole	China	Clinical	[86,90]
		T315A, Y118A, Y18F and Y118T	Non-synonymous substitution	Fluconazole	-	Lab-created	[91,92]
		K128T	Non-synonymous substitution	Likely no effect	China	Clinical	[86,93]
		D116E and E266D	Non-synonymous substitution	No effect on protein function or resistance	USA	Clinical	[36]
	*C. auris*	F126T, Y132F and K143R	Non-synonymous substitution	Fluconazole	South Africa, Venezuela, India	Clinical	[28]
	*N. glabrata*	C108G, C423T and A1581G	Synonymous substitution	No effect	Brazil	Clinical	[94]
		T768C, A1023G and T1557A	Synonymous substitution	No effect	Slovakia	Clinical	[95]
		E502V	Non-synonymous substitution	No effect	Slovakia	Clinical	[96]
	*P. kudriavzevii*	G524R	Non-synonymous substitution	No effect on protein function or resistance	Brazil	Clinical	[94]
		Y166S	Non-synonymous substitution	Voriconazole	Brazil	Clinical	[94]
	*C. tropicalis*	Y132F	Missense	Fluconazole	Brazil	Clinical	[97]
		K143R	Non-synonymous substitution	Fluconazole, voriconazole and itraconazole	Brazil	Clinical	[98]
**ERG3** (C5 sterol desaturase)	*N. glabrata*	Q139A	Non-synonymous substitution	Fluconazole	Korea	Clinical	[99]
**UPC2** (TF, regulates most ERG genes)	*C. albicans*	G648D, G648S, A643T, Y642F, A646V and W478C	GOF substitution	Fluconazole	USA	Clinical	[100]
		A643V	GOF substitution	Fluconazole	USA	Clinical	[100]
		G307S and G448E	GOF substitution	Fluconazole	Germany	Clinical	[101]

**Table 3 cells-12-02655-t003:** Mutations in cell membrane genes in pathogenic *Candida* species implicated in antifungal resistance.

Gene	*Candida* Species	Mutation	Type of Mutation	Antifungal Resistance	Location	Isolate Type	Ref.
**CDR1 + CDR2** (ABC-Ts)	*C. albicans*	Chr 3 trisomy	Increased *cdr1* and *cdr2* copy numbers	Azole	-	In vitro	[124]
**MLT1** (ABC-T)	*C. albicans*	K710A	Loss of function	Reduced azole resistance	-	In vitro	[123]
		F765Δ	Loss of function	Reduced azole resistance	-	In vitro	[123]
**TAC1** (TF, regulates CDR1, CDR2 and PDR16)	*C. albicans*	T225A, V736A, N972D, N977D, G980E and G980W	GOF substitution	Azole	USA	Clinical	[125]
	*C. auris*	K143R, F214S, R495G and A640V	Non-synonymous substitution	Fluconazole	USA	Clinical/ in vitro	[126]
**MRR1** (TF, regulates MDR1)	*C. albicans*	P683S and P683H	GOF substitution	Azole	Germany	Clinical	[101,127]
	*C. dubliniensis*	T374I, S595Y and C866Y	GOF substitution	Azole	Ireland	Clinical	[128,129,130]
		T965∆ and (D987-I998)∆	Deletion	Azole	Ireland	Clinical	[128]
**PDR16** (phosphatidylinositol transfer protein)	*N. glabrata*	∆pdr16	Gene deletion	Reduced resistance to fluconazole, itraconazole and ketoconazole miconazole	*-*	In vitro	[131]
**PDR1** (TF, regulates CDR1, SNQ2, PDH1 and QDR2)	*N. glabrata*	Hotspot regions: 312–382, 800–1107 and 539–632	GOF substitution	Azole	Italy, Switzerland, France and Japan	Clinical	[35,132]
		L328F, R376W, D1082G, T588A, T607S, E1083Q, Y584C, D876Y, L280F, N691D, S316I, D261G, R293I, R592S, G583S, S343F and R376G	GOF substitution	Fluconazole	Italy, Switzerland, France and Japan	Clinical	[132]

**Table 4 cells-12-02655-t004:** Mutations in cell wall genes in pathogenic *Candida* species implicated in antifungal resistance.

Gene	*Candida* Species	Mutation	Type of Mutation	Antifungal Resistance	Location	Isolate Type	Ref.
**FKS1** *(*β1–3 glucan synthase)	*C. albicans*	Hotspot regions: aa 637–654 and 1345–1365	Non-synonymous substitution	Echinocandin	-	Clinical	[74,162]
		S645F	Non-synonymous substitution	Echinocandin	USA	Clinical	[170]
	*C. auris*	F635Y, F635L, S639F and R1354S	Non-synonymous substitution	Echinocandin	India	In vitro/ in vivo	[168]
	*N. glabrata*	F625C and S629P	Non-synonymous substitution	Echinocandin	-	Clinical/ in vitro	[171,172]
		F625∆	Deletion	Echinocandin	-	Clinical/ in vitro	[171,172]
	*P. kudriavzevii*	F655C	Non-synonymous substitution	Echinocandin	USA	Clinical	[166]
	*C. parapsilosis*	P660A	Non-synonymous substitution	Echinocandin	-	All species members	[161]
**FKS2** (β1–3 glucan synthase)	*N. glabrata*	F659S and F659V	Non-synonymous substitution	Echinocandin	USA	Clinical	[158,159,173]
		F659∆	Deletion	Echinocandin	USA	Clinical	[158,159,173]
		S663P and S663F	Non-synonymous substitution	Echinocandin	USA	Clinical	[171,172]
		E655G, E655K, P667H and P667T	Non-synonymous substitution	Echinocandin	USA	Clinical	[171,172]
		R1378S and R1378G	Non-synonymous substitution	Echinocandin	USA	Clinical	[171,172]

**Table 5 cells-12-02655-t005:** Mutations in nucleic acid biosynthesis genes in pathogenic *Candida* species implicated in antifungal resistance.

Gene	*Candida* Species	Mutation	Type of Mutation	Antifungal Resistance	Location	Isolate Type	Ref.
**FCA1/FCY1** (cytosine deaminase)	*C. albicans*	G28D and S29L	LOF substitution	5FC	UK	Clinical	[197]
	*C. dubliniensis*	S29L	Non-synonymous substitution	5FC	Egypt and Saudi Arabia	Clinical	[190]
	*N. glabrata*	A15D, G11D and W148R	Non-synonymous substitution	5FC	-	In vitro	[201]
**FCY2** (cytosine permease)	*C. albicans*	A176G	LOF substitution	5FC	UK	Clinical	[197]
	*C. tropicalis*	G145T	Non-synonymous substitution	5-FC	Taiwan	Clinical	[200]
**FUR1** (uracil phosphoribosyltransferase (UPRT))	*C. albicans*	C101R	LOF substitution	5FC	Multiple countries	Clinical	[196,197]
	*N. glabrata*	G190D	LOF substitution	5FC	France	Clinical	[195]
		I83K and D193G	LOF substitution	5FC/5FU	-	In vitro	[201,202]
		∆G73-V81	LOF Deletion	5FC/5FU	-	In vitro	[201,202]
**MSH2** (DNA mismatch repair)	*N. glabrata*	V239L	Non-synonymous substitution	Fluconazole or echinocandin	Multiple countries	Clinical	[35]

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
