# Peer review of "Molecular Mechanisms Associated with Antifungal Resistance in Pathogenic Candida Species"

_cells, 2023, doi:10.3390/cells12222655_

Round 1

Reviewer 1 Report

Comments and Suggestions for Authors

Manuscript titled "Molecular mechanisms associated with anti fungal resistance in pathogenic Candida species" by Karolina et al is a nice piece of writing. The review article is quite informative and comprehensive. Overall the manuscript is OK just need minor revision before can be accepted for publication in journal. My comments and suggestions to authors are given below.

1) Minor writing check throughout the draft.

2) Author mentioned different candida species known to infect humans, it will be great if author can make table just highlighting different candida species known to infect humans, there spread, severity. This will increase the importance of draft.

3) In future direction author should also briefly discuss the ways to overcome rise in anti fungal resistance which is some thing applicable to other fungal species.

4) People in community is talking about use or development of anti fungal vaccine as a means to tackle anti-fungal resistance (see recent review https://www.frontiersin.org/articles/10.3389/ffunb.2023.1241539/full), author should discuss this also or include important reference.

5)The factor increasing fungal infection an hence rise in anti fungal resistance should be discussed or include proper reference

6) Since the genes/proteins mentioned in table are conserved in fungal cells, whether other species also showed anti-fungal resistance (involving same gene/proteins should be mentioned briefly). This will make the importance more broader.

7) Where in table author mentioned gene (just point out whether those are conserved in other fungal species)

8) If possible, in introduction author should mention  importance of fungal infection (in terms of loss of life, economic etc). This is to give background for the review.

Comments on the Quality of English Language

Overall writing is OK

Author Response

Please see attached letter.

Reviewer 2 Report

Comments and Suggestions for Authors

The primary aim of this timely review was to summarise the molecular mechanisms associated with antifungal resistance in Candida species. The review provides an excellent overview of the known gene mutations for the main, commercially available antifungals used for yeast infections. The review is clearly written, the figures are informative and beautifully illustrated, and the standard of English is excellent. As such this review is a very good source of information for readers of all abilities and levels of mycological knowledge.

The background information provided is sufficient to understand the context of the more detailed descriptions of the molecular mechanisms of antifungal drug resistance, which is the main point of this review. In addition to noting the genetic markers for potential development of resistance testing, it would have been useful for the reviewers to summarise the kits that are commercially available and reflect on their efficacy by including any relevant studies. Although efforts were made to illuminate potential molecular drug targets for the development of research-based tests, provision of a summary of commercial tests may have extended the readership to clinical scientists.

Nonetheless, this review is worthy of publication. Several minor points are provided by this reviewer as an attachment for the attention and consideration by the authors and copy editors to improve the review in advance of publication.

Please address the reviewer's comments in the attached pdf file.

Author Response

Please see attached letter.

Reviewer 3 Report

Comments and Suggestions for Authors

Dear authors,

Please consider the following observations:

On line 21 of the abstract ERG11, UPC2 preferably in capital letters, same with the name of the other genes cdr1, cdr2, mdr1, pdr16, snq2, (fks1, fks2 and pdr1) (fcy1, fcy2, fur1), and preferably change it throughout the document

In lines 39 and 40 after mucosal tissues such as the oral cavity, can be completed with gastrointestinal mucosa and vaginal mucosa.

On line 48 remove the space between the references and the word Numerous

In point 1.2.1. Candida in italics

In line 77 Candida in italics

In point 1.2.2. Candida auris in italics

In Table 1 there is no data on C. glabrata and C. krusei with Posaconazole and Itraconazole?

The chemical formulas in figure two do not have good definition

In figure two in the echinocandins and nucleoside analog sections, Candida should be italicized

In section 2.1 (azoles), include in the first paragraph that one of the characteristics is the low absorption of some azoles such as itraconazole as well as their difficult dilution.

In session 2.2 (Polyenes) highlight the percentages of resistance of C. auris to Amphotericin B

In section 5.1 Fks1 and fks2 sequence mutations include the mutations described in the FKS1 resistance gene in C. auris (F635Y, F635L, S639F and R1354S) and include this information in table 4.

At the end of section 7. Biofilm formation and antifungal resistance, include C. auris, this species also has the capacity to form biofilms, not as strong as C. albicans but it does produce them.

On line 581 Candida in italics

Author Response

Please see attached letter.
